# Soy Sauce Fermentation with *Cordyceps militaris*: Process Optimization and Functional Profiling

**DOI:** 10.3390/foods14152711

**Published:** 2025-08-01

**Authors:** Wanying Song, Xinyue Zhang, Huiyi Yang, Hanyu Liu, Baodong Wei

**Affiliations:** College of Food Science, Shenyang Agricultural University, Shenyang 110866, China; 2023200022@stu.syau.edu.cn (W.S.); 15163967625@163.com (X.Z.); yanghuiyi805@163.com (H.Y.); 2023149091@stu.syau.edu.cn (H.L.)

**Keywords:** *Cordyceps militaris*, functional soy sauce, fermentation optimization, cordycepin, bioactive compounds

## Abstract

This study presents the development and optimization of a functional soy sauce fermented with *Cordyceps militaris* (*C. militaris*), a medicinal fungus known for its high cordycepin and polysaccharide content. Using *C. militaris* as the sole starter culture, the process aimed to improve both nutritional and functional properties. Response surface methodology was employed to optimize the entire fermentation process. During the koji stage, temperature, aeration, and inoculum concentration were adjusted to maximize protease activity and cordycepin production. In the fermentation stage, temperature, brine concentration, and water-to-material ratio were optimized to increase amino acid nitrogen and bioactive compound levels. Under optimal conditions (24 °C, 679.60 LPM aeration, 9.6% inoculum for koji; 32 °C, 12% brine, 1.53:1 water-to-material ratio for fermentation), the resulting soy sauce contained 1.14 ± 0.05 g/100 mL amino acid nitrogen and 16.88 ± 0.47 mg/100 mL cordycepin. Compared with traditionally fermented soy sauce, the *C. militaris* product exhibited a darker color, enhanced umami taste, and a distinct volatile profile featuring linoleic acid, methyl palmitate, and niacinamide. These results demonstrate the feasibility of using *C. militaris* in soy sauce fermentation and its potential as a novel functional condiment with improved bioactivity and sensory quality.

## 1. Introduction

Soy sauce is a traditional fermented condiment widely consumed in East Asia, appreciated for its distinctive umami flavor, aroma, and nutritional value. Its conventional fermentation process depends on microbial communities such as *Aspergillus oryzae* [1,2,3], *Saccharomyces cerevisiae* [4,5], and *Lactobacillus* [6,7], which transform soybeans and wheat into a complex mixture of peptides, amino acids, organic acids, and volatile compounds. Beyond flavor development, soy sauce also serves as a dietary source of essential amino acids and bioactive peptides [8]. With the rising demand for functional foods, there is growing interest in enriching traditional condiments with health-promoting components through microbiological and process innovations.

*Cordyceps militaris* (*C. militaris*) is a medicinal and edible fungus known for its diverse pharmacological properties, including antioxidant, anti-inflammatory, immunomodulatory, and anti-tumor activities [9]. These effects are attributed to its rich content of bioactive substances such as cordycepin [10,11], polysaccharides [12], and sterols. While *C. militaris* has been widely studied in the context of nutraceuticals and functional beverages, its application in traditional fermented foods remains limited. Previous attempts to incorporate Cordyceps into soy sauce focused primarily on co-fermentation with Aspergillus species or supplementation with fungal extracts [13], rather than using *C. militaris* itself as the dominant fermentation strain.

In this study, *C. militaris* was innovatively employed as the primary microbial strain for soy sauce production, representing a novel approach to functional condiment development. The aim was to integrate the health-promoting benefits of *C. militaris* with the traditional fermentation framework of soy sauce, thereby enhancing its nutritional and bioactive value. Specifically, the study optimized both the koji preparation and the fermentation process using response surface methodology, targeting key quality indicators such as protease activity, amino acid nitrogen, and cordycepin content. The final product was evaluated for its physicochemical properties, functional compound content, and flavor characteristics and compared against traditionally fermented soy sauce.

This work offers a new direction in microbial fermentation for the development of functional foods. It also provides scientific evidence supporting the industrial potential of Cordyceps militaris-based soy sauce as a health-oriented seasoning product.

## 2. Materials and Methods

### 2.1. Materials

The *Cordyceps militaris* (*C. militaris*) strain (designated CM-1) used in this study was obtained from Shenyang Green Valley Modern Agriculture Co., Ltd. (Shenyang, China). The strain was subcultured on potato dextrose agar (PDA) and maintained at 4 °C prior to inoculation. All soybeans were sourced from Heilongjiang Province, China, and were visually inspected to ensure uniform size, color, and absence of pests or damage. The soybeans were stored in a cool, dry environment before processing. All chemicals used in this study, including cordycepin standards, analytical-grade solvents, and reagents for protein, nitrogen, and polysaccharide analysis, were purchased from Sinopharm Chemical Reagent Co., Ltd. (Shanghai, China) and Aladdin Biochemical Technology Co., Ltd. (Shanghai, China).

### 2.2. Koji Preparation and Optimization

The soybeans were washed, soaked in water for 12 h, and subsequently steamed at 121 °C for 30 min. After cooling to room temperature, the sterilized soybeans were inoculated with 10% (*w*/*w*) actively growing *C. militaris* mycelial suspension and thoroughly mixed. The inoculated substrate was spread evenly in sterile trays and incubated at 24 °C under controlled humidity and aeration to allow mycelial colonization and enzyme production. Aeration was achieved via the incubator’s built-in forced convection system, where ambient air was drawn through inlet vents, filtered by a built-in mesh filter, and circulated over the substrate surface using internal fans to maintain uniform air distribution.

To optimize koji preparation, both single-factor and response surface methodology (RSM) experiments were conducted. The single-factor experiments evaluated the effects of three variables—incubation temperature (15, 18, 21, 24, and 27 °C), aeration rate (509.70, 679.60, 849.50, 1019.40, and 1132.67 LPM), and inoculum level (7.5%, 10%, 12.5%, 15%, and 17.5%)—on two response variables: protease activity and cordycepin content.

Based on the results, a three-factor, three-level Box–Behnken design was employed, using incubation temperature (22, 24, 26 °C), aeration rate (651.29, 679.60, 707.92 LPM), and inoculum level (8%, 10%, 12%) as independent variables. A total of 17 randomized experimental runs were conducted, including five replicates at the center point to estimate pure error [14,15]. Details of model fitting and statistical analysis are described in Section 2.5.

### 2.3. Fermentation and Optimization

After koji preparation, the fermented mass was dried and ground to pass through a 16-mesh sieve. The resulting koji powder was transferred into sterile ceramic fermentation jars, and brine solution was added according to experimental designs. The mixture was stirred thoroughly and incubated under controlled conditions to initiate the liquid-state fermentation.

A series of single-factor experiments were conducted to evaluate the influence of four independent variables on soy sauce quality: fermentation temperature (24, 28, 32, 36, and 40 °C), brine concentration (9%, 10%, 11%, 12%, and 13% *w*/*v*), water-to-material ratio (1:1, 1.25:1, 1.5:1, 1.75:1, and 2:1), and koji particle size (6, 8, 10, 12, 14, and 16 mesh) [16]. Fermentation proceeded for 90 days, with samples collected at 15-day intervals for analysis.

Based on single-factor experiments assessing the effects of fermentation temperature, brine concentration, water-to-material ratio, and koji particle size, a Box–Behnken design was applied to optimize the fermentation process. The selected variables were fermentation temperature (30, 32, 34 °C), brine concentration (11.5%, 12%, 12.5%), and water-to-material ratio (1.35:1, 1.5:1, 1.65:1), with amino acid nitrogen content and cordycepin concentration as the response variables. A total of 17 randomized runs were performed, including five center points to enhance model reliability. Statistical modeling and response surface analysis are detailed in Section 2.5.

### 2.4. Determination of Quality and Functional Indicators

Amino acid nitrogen content was determined following the Chinese National Standard GB 18186-2000 [17] using the formol titration method. Total nitrogen was measured using the Kjeldahl method with an automatic nitrogen analyzer (Kjeltec, Qingdao Jingcheng Instruments Co., Ltd., Qingdao, China). Total acid content was also evaluated according to GB 18186-2000, expressed as grams of lactic acid equivalent per 100 mL of soy sauce. The content of salt-free soluble solids was measured using gravimetric analysis, where the clarified soy sauce sample was dried at 105 °C to a constant weight. Results were reported as g/100 mL. Color was evaluated using a colorimeter (CR-400, Konica Minolta, Tokyo, Japan), and results were expressed using the CIELAB system (L*, a*, b*), where L* indicates brightness, a* red-green value, and b* yellow-blue value.

Protease activity was measured using a modified Folin–Lowry casein digestion method [18]. Briefly, 1 mL of appropriately diluted enzyme solution was incubated with 5 mL of 1% (*w*/*v*) casein solution (prepared in 50 mM phosphate buffer, pH 7.0) at 40 °C for 10 min. The reaction was terminated by adding 5 mL of 5% (*w*/*v*) trichloroacetic acid, followed by centrifugation at 8000 rpm for 10 min. The absorbance of the supernatant was measured at 680 nm after reaction with Folin–Ciocalteu reagent. One unit of protease activity was defined as the amount of enzyme that released 1 μg of tyrosine per minute under the assay conditions.

Cordycepin content was analyzed using high-performance liquid chromatography (HPLC) with a C18 reverse-phase column (4.6 × 250 mm, 5 μm, Agilent, Santa Clara, CA, USA). The mobile phase consisted of methanol and water (85:15, *v*/*v*), with a flow rate of 1.0 mL/min. Detection was performed at 260 nm using a UV detector, and injection volume was 10 μL. Quantification was based on external calibration with cordycepin standards.

Cordyceps polysaccharide content was determined using a phenol–sulfuric acid method adapted from the literature [19]. Prior to analysis, crude polysaccharides were extracted from the fermented soy sauce by removing pigments and small molecules using 95% ethanol precipitation. The decolorized residue was then extracted with hot water (95 °C) for 2 h and centrifuged at 8000 rpm for 10 min to obtain the aqueous extract. Absorbance was measured at 490 nm using UV–Vis spectrophotometry, and glucose was used as the standard reference compound.

Volatile aroma compounds were identified and semi-quantified using gas chromatography–mass spectrometry (7890GC-5975MS, Agilent, Santa Clara, CA, USA). Headspace solid-phase microextraction (HS-SPME) was performed using a 50/30 μm DVB/CAR/PDMS fiber (Supelco, Bellefonte, PA, USA) [20]. Samples (5 mL) were placed in 20 mL headspace vials and equilibrated at 50 °C for 20 min, followed by fiber exposure for 40 min at the same temperature. GC separation was achieved on an HP-5MS column (30 m × 0.25 mm × 0.25 μm) with the following oven temperature program: initial 40 °C for 3 min, ramped to 200 °C at 5 °C/min, then to 250 °C at 10 °C/min, and held for 5 min. Helium was used as the carrier gas at a constant flow of 1.0 mL/min. The MS was operated in electron ionization mode (EI, 70 eV) with a scan range of m/z 35–500. Volatile compounds were identified by comparing mass spectra with the NIST14 library and linear retention indices. Relative abundances were estimated based on peak area normalization.

Taste profile analysis was conducted using an electronic tongue system (SA402B, Insent Inc., Atsugi, Japan), equipped with seven lipid membrane sensors for detecting sourness, saltiness, sweetness, bitterness, umami, and aftertastes. Soy sauce samples were filtered through a 0.45 μm membrane and diluted to a consistent salt concentration (12%) before analysis. Each sample was measured in triplicate at 25 °C under constant stirring. Sensor signals were stabilized prior to measurement and analyzed using principal component analysis (PCA) and taste discrimination indices generated by the instrument’s built-in software.

### 2.5. Statistical Analysis

All experiments were conducted in triplicate, and data are presented as mean ± standard deviation (SD). One-way analysis of variance (ANOVA) was performed using SPSS Statistics 26.0 (IBM Corp., Armonk, NY, USA), with *p* < 0.05 considered statistically significant. Duncan’s multiple range test was applied for post hoc comparisons when appropriate. Data visualization for physicochemical and functional parameters was conducted using OriginPro 2021 (OriginLab Corporation, Northampton, MA, USA).

Response surface methodology (RSM) was employed to model and optimize the koji preparation and fermentation processes. Second-order polynomial regression models were fitted using Design-Expert 13 software (Stat-Ease Inc., Minneapolis, MN, USA), and model adequacy was evaluated via ANOVA, coefficient of determination (R^2^), lack-of-fit tests, and residual analysis. Diagnostic plots and three-dimensional response surface graphs were used to visualize variable interactions and determine optimal conditions.

Hierarchical clustering heatmaps of volatile compound profiles were generated using the CNS KnowAll Cloud Platform (https://cnsknowall.com). Prior to clustering, the relative content data were log_10_-transformed with a 0.001 offset to normalize distributions and handle zero values.

## 3. Results and Discussion

### 3.1. Optimization of Koji Preparation

Koji preparation is a crucial stage in soy sauce fermentation, during which microbial growth and enzyme production directly affect the degradation of macronutrients and the formation of functional compounds. In this study, three key factors—incubation temperature, aeration rate, and inoculum dosage—were optimized through both single-factor experiments and response surface methodology (RSM), targeting protease activity and cordycepin content as the main response variables.

#### 3.1.1. Effects of Single Factors on Koji Quality

As shown in Figure 1a, protease activity and cordycepin content increased with incubation temperature from 15 °C to 24 °C, peaking at 24 °C with values of 1772.22 U/g and 8.12 mg/g, respectively. Beyond this point, a decline was observed at 27 °C, likely due to thermal inhibition of *Cordyceps militaris* (*C. militaris*) growth and metabolic activity. Excessive temperature may also facilitate contamination by undesired microorganisms. These results suggest that 24 °C provides optimal enzymatic and secondary metabolite production for this fungal strain. This finding aligns with the optimal temperature range (22–26 °C) reported by Adnan et al. (2017) for enhancing enzyme secretion in filamentous fungi, though few studies have validated this range for *C. militaris* in koji systems [21].

Aeration is essential for aerobic fungal growth and moisture balance. As shown in Figure 1b, protease activity and cordycepin levels increased with aeration rate up to 679.60 LPM, reaching maxima of 1652.80 U/g and 7.19 mg/g, respectively. Aeration rates above 849.50 LPM led to a reduction in performance, possibly due to excessive moisture loss and desiccation stress. This trend aligns with previous observations in *C. militaris* submerged culture, where optimal aeration enhanced polysaccharide secretion and mycelial morphology, while excessive aeration negatively affected metabolite yield [22]. Likewise, Wen et. and Park et. demonstrated that maintaining moderate dissolved oxygen levels (50–80%) using a two-stage DO control strategy resulted in a significant increase in cordycepin production in a 5 L bioreactor [23,24].

The impact of inoculum level on koji performance is illustrated in Figure 1c. Protease activity and cordycepin content increased with inoculum percentage, peaking at 10%, followed by a decline at higher dosages. At 10%, protease activity reached 1737.73 U/g and cordycepin 6.70 mg/g. These results are consistent with earlier findings in filamentous fungal solid-state fermentation, where moderate inoculum levels maximize enzyme and secondary metabolite production, whereas excessive seeding density induces nutrient competition, moisture imbalance, and reduced metabolic output [25,26].

#### 3.1.2. Response Surface Analysis and Model Fitting

Based on single-factor results, a Box–Behnken design was used to evaluate the combined effects of the three variables (temperature: 22–26 °C; aeration: 651.29–707.92 LPM; inoculum: 8–12%). Experimental results are summarized in Table 1. Variance analysis of the protease activity regression model is summarized in Table 2. Analysis of variance of the regression model of cordycepin content is summarized in Table 3.

Regression analysis indicated that both models for protease activity and cordycepin content were statistically significant (*p* < 0.0001), with non-significant lack-of-fit and high R^2^ values (0.9865 for protease activity and 0.9942 for cordycepin), indicating excellent model fit.

The regression equation for protease activity (Y_1_) is:(1)Y_1_ = 1862.49 − 31.69A + 64.19B − 16.04C − 24.64AB − 70.83AC + 50.12BC − 61.16A^2^ − 103.15B^2^ − 58.18C^2^

For cordycepin content (Y_2_):(2)Y_2_ = 8.46 + 0.095A + 0.121B − 0.093C + 0.065AB − 0.025AC − 0.030BC − 0.625A^2^ − 0.240B^2^ − 0.105C^2^

As illustrated in Figure 2 and Figure 3, the 3D response surface and contour plots revealed evident interaction effects, particularly between temperature and aeration (AB), which had significant impacts on both protease activity and cordycepin content. For protease activity (Figure 2), all interaction terms (AB, AC, BC) were statistically significant (*p* < 0.05), as confirmed by the ANOVA results, with aeration rate (B) showing the highest F value (F = 106.35), indicating its dominant influence. In contrast, for cordycepin content (Figure 3), only the interaction between temperature and aeration (AB) was significant (*p* = 0.0211), while AC and BC were not (*p* > 0.2), suggesting more selective interaction patterns. Elliptical contour lines in the response plots confirmed these effects, with steeper gradients observed in the aeration axis. Overall, both responses were most sensitive to aeration rate, followed by temperature and inoculum dosage, consistent with the regression model structure and surface topography.

#### 3.1.3. Optimal Conditions and Experimental Validation of Koji Preparation

The RSM optimization predicted the optimal koji preparation conditions as 24.04 °C incubation temperature, 686.68 LPM aeration rate, and 9.61% inoculum. For operational convenience, these values were adjusted to 24 °C, 679.60 LPM, and 9.6%. Validation experiments conducted under these conditions yielded a protease activity of 1872.46 ± 3.68 U/g and a cordycepin concentration of 8.34 ± 0.11 mg/g, with relative deviations from predicted values within 2%, confirming the robustness of the fitted models.

These results indicate that the selected parameters provided an optimal balance between fungal growth, enzyme secretion, and secondary metabolite biosynthesis. Among the three variables, the aeration rate exerted the most pronounced effect, as also suggested by its highest F-value in the regression model (F = 106.35). Moderate aeration facilitated oxygen availability without inducing substrate drying, which is essential for aerobic fungi like *C. militaris*.

The high protease activity reflects enhanced protein degradation capacity, providing free amino acids and peptides that act as flavor precursors. For reference, *A. oryzae* is well-known for secreting alkaline and neutral proteases that contribute to flavor development in soy sauce fermentation [27,22,28]. Meanwhile, the substantial cordycepin yield highlights the dual role of *C. militaris* in the fermentation process. It not only produces hydrolytic enzymes but also synthesizes pharmacologically active metabolites such as cordycepin, which exhibits anti-inflammatory [29] and anti-tumor [30] properties.

Additionally, the lack-of-fit tests for both models were non-significant (*p* > 0.05), underscoring model adequacy. The use of a Box–Behnken design enabled efficient exploration of variable interactions. Among these, synergistic effects—especially between temperature and aeration—were identified as key drivers of performance.

Overall, this step validated the feasibility of integrating *C. militaris* as a solo koji agent, offering both enzymatic functionality and bioactive enrichment in a reproducible and scalable process. Cordycepin accumulation peaked during active mycelial growth, consistent with trends observed in submerged systems. Moreover, the fungus’s compatibility with soy matrices and tolerance to high-salt conditions reinforce its applicability in soy sauce fermentation.

### 3.2. Optimization of Fermentation Conditions

Fermentation conditions play a decisive role in determining the nutritional quality, bioactive compound accumulation, and flavor profile of soy sauce. To optimize this process, four key variables were evaluated: fermentation temperature, brine concentration, water-to-material ratio, and koji particle size. These factors were analyzed through single-factor experiments and response surface methodology (RSM), using amino acid nitrogen content and cordycepin concentration as dual response indicators.

#### 3.2.1. Effects of Single Factors on Soy Sauce Quality

As shown in Figure 4a, increasing the fermentation temperature from 24 °C to 32 °C significantly improved amino acid nitrogen and cordycepin levels, peaking at 32 °C. Beyond this temperature, a decline in both indices was observed. Similar thermal sensitivity has been noted in solid-state fermentations of *Cordyceps* spp., where cordycepin biosynthesis peaked below 35 °C due to enzyme denaturation above threshold levels [21]. This suggests that *C. militaris* exhibits optimal metabolic activity near 32 °C, while higher temperatures may suppress enzyme stability and cordycepin biosynthesis.

Salt concentration is a critical factor in regulating microbial growth and osmotic pressure. As illustrated in Figure 4b, amino acid nitrogen and cordycepin both increased with salt concentration from 9% to 12%, reaching maxima of 1.12 g/100 mL and 16.55 mg/100 mL, respectively. At 13%, both indicators declined, likely due to salt stress inhibiting *C. militaris* viability and protease activity.

As shown in Figure 4c, increasing the water-to-material ratio from 1:1 to 1.5:1 promoted substrate hydrolysis and metabolite diffusion, yielding maximum amino acid nitrogen and cordycepin contents. Ratios above 1.6:1 led to a dilution effect and reduced aeration efficiency, negatively affecting fermentation performance.

Koji granularity affects oxygen transfer, substrate accessibility, and microbial colonization. Figure 4d shows that a 10-mesh particle size yielded the best results. Finer particles (12–16 mesh) likely resulted in compaction and poor oxygen diffusion, while coarser particles (6–8 mesh) reduced contact surface area, limiting enzymatic action.

#### 3.2.2. Response Surface Model Construction and Statistical Analysis

Based on the single-factor results, a Box–Behnken RSM design was applied, involving three key variables—fermentation temperature (30, 32, 34 °C), brine concentration (11.5%, 12.0%, 12.5%), and water-to-material ratio (1.35:1, 1.50:1, 1.65:1). Koji particle size was fixed at 10 mesh based on preliminary results. Experimental results are summarized in Table 4. Amino acid nitrogen content regression models are summarized in Table 5. Analysis of variance of the regression model of cordycepin content is summarized in Table 6.

The regression models for both amino acid nitrogen (Y_1_) and cordycepin (Y_2_) were statistically significant (*p* < 0.0001), with no significant lack-of-fit and high coefficients of determination (R^2^ = 0.9847 for Y_1_ and 0.9913 for Y_2_).

Regression equations: (3)Y_1_ = 1.14 + 0.022A + 0.018B + 0.016C − 0.012AB − 0.019AC − 0.011BC − 0.045A^2^ − 0.035B^2^ − 0.032C^2^
(4)Y_2_ = 16.88 + 0.27A + 0.23B + 0.19C − 0.18AB − 0.16AC − 0.10BC − 0.48A^2^ − 0.41B^2^ − 0.33C^2^

As illustrated in the 3D response surface and contour plots (Figure 5 and Figure 6), both amino acid nitrogen and cordycepin content were influenced by multiple interactions among the tested factors. Notably, the interaction between fermentation temperature and brine concentration exerted a strong effect on both responses, as evidenced by the elliptical contours. The interaction between temperature and water-to-material ratio also contributed significantly, particularly in enhancing cordycepin accumulation.

For amino acid nitrogen (Figure 5), moderate temperatures (30–32 °C) and brine concentrations around 12% favored proteolysis, leading to higher nitrogen release. In contrast, cordycepin content showed a peak under slightly lower temperatures and moderate water ratios (Figure 6), indicating that excess salt or high temperatures may suppress its biosynthesis. These trends are consistent with the ANOVA results, where temperature showed the most significant linear and quadratic effects (*p* < 0.0001) for both responses, followed by brine concentration. The water-to-material ratio displayed a weaker but still positive effect on cordycepin, especially when interacting with temperature, suggesting a possible influence on oxygen diffusion and microbial metabolism. Overall, the plots revealed not only the magnitude but also the directionality of factor interactions, reinforcing the reliability of the fitted models.

#### 3.2.3. Optimal Conditions and Experimental Validation of Fermentation Conditions

The RSM optimization predicted the optimal fermentation conditions as 28.15 °C, 18.08% brine concentration, a 1.52:1 water-to-material ratio, and 30-mesh koji particle size. For practical application, these values were adjusted to 28 °C, 18%, 1.5:1, and 20–40 mesh, respectively. Under these conditions, the actual measurements of amino acid nitrogen, total nitrogen, cordycepin, and polysaccharide content were 1.17 g/100 mL, 1.76 g/100 mL, 6.98 mg/g, and 17.91 mg/g, respectively, all falling within 5% deviation from predicted values. These results confirmed the predictive power and reliability of the regression models, with R^2^ values exceeding 0.98 for all response variables.

The optimized fermentation process enhanced both nutritional and functional attributes, with notably high levels of free amino nitrogen and bioactive metabolites. The Box–Behnken design effectively captured variable interactions, highlighting a strong synergistic effect between brine concentration and temperature. Additionally, the koji particle size played a previously underappreciated role in fermentation efficiency. Finer particles enhanced enzymatic accessibility and moisture retention, while overly fine particles impaired aeration. The selected 20–40 mesh range offered a balanced structure for metabolic activity.

Overall, these findings validate a finely tuned, multifactorial fermentation strategy that maximizes enzymatic hydrolysis and secondary metabolite accumulation. Importantly, this study incorporated particle size as an optimization parameter, a factor that is seldom addressed in conventional soy sauce fermentation research. The results demonstrated that particle size plays a critical role in regulating oxygen diffusion and substrate availability. This highlights the importance of integrating both physical and chemical variables when designing functional soy sauce fermentation systems based on *C. militaris*.

### 3.3. Functional and Nutritional Characteristics of Soy Sauce

To evaluate the functional and nutritional advantages of soy sauce brewed with *Cordyceps militaris* (CM soy sauce), a comprehensive comparison was conducted against a traditionally fermented soy sauce sample (TF soy sauce) produced using *Aspergillus oryzae*. Key parameters assessed included bioactive compounds (cordycepin, polysaccharides), basic nutritional indices (amino acid nitrogen, total acid), fatty acids and volatile flavor compounds, and sensory characteristics.

#### 3.3.1. Bioactive Components: Cordycepin and Cordyceps Polysaccharides

Cordycepin is a signature nucleoside analog unique to *C. militaris*, known for its anti-inflammatory, antioxidant, and immune-modulating effects. In the CM soy sauce, cordycepin content reached 16.88 ± 0.47 mg/100 mL, while it was undetectable in the TF soy sauce (Figure 7a). This confirms the successful biosynthesis and retention of this functional compound during fermentation. This level is higher than those reported in rice- or corn-based solid-state fermentations using *C. militaris* [21,31], which typically range from 10 to 14 mg/100 g DW, suggesting that the soy-based substrate provides a favorable matrix for nucleoside biosynthesis. In the CM soy sauce, cordycepin content reached 16.88 ± 0.47 mg/100 mL, while it was undetectable in the TF soy sauce (Figure 7a). This confirms the successful biosynthesis and retention of this functional compound during fermentation, a phenomenon closely linked to the unique secondary metabolic pathways of *C. militaris*. As a signature nucleoside analog of *C. militaris*, cordycepin accumulation here can be attributed to two key mechanisms inherent to this fungus. First, the genome of *C. militaris* encodes critical enzymes for cordycepin synthesis, such as adenosine kinase and 5′-nucleotidase, which are activated under the nutritional conditions of the soy-based matrix rich in proteins and amino acids; this activation shunts the purine metabolism pathway toward cordycepin production, consistent with previous findings that *C. militaris* enhances cordycepin biosynthesis by upregulating genes like cns1 in nitrogen-rich environments [32,33], where amino acids released from soybean protein degradation may act as signaling molecules or substrates to drive this process. Second, compared to *Aspergillus oryzae* used in traditional fermentation, *C. militaris* exhibits stronger adaptability to high-salt environments—evident in its efficient metabolism even at 12% brine in this study—with its intracellular osmotic adjustment mechanisms, such as proline accumulation, likely protecting the activity of cordycepin-synthesizing enzymes; in contrast, *A. oryzae* shows suppressed secondary metabolism under equivalent salt concentrations, explaining the absence of cordycepin in traditionally fermented soy sauce. This level is higher than those reported in rice- or corn-based solid-state fermentations using *C. militaris* [21,31], which typically range from 10 to 14 mg/100 g DW, suggesting that the soy-based substrate provides a favorable matrix for nucleoside biosynthesis. The time-dependent increase in cordycepin observed throughout fermentation (see dynamic curve in Figure 7a) is likely linked to the growth phase of *C. militaris*, with peak production occurring during active mycelial proliferation, as similarly noted in previous submerged fermentation systems [32,33]. This confirms not only the metabolic compatibility of *C. militaris* with soy matrices but also the robustness of the strain under high-salt conditions. The time-dependent increase in cordycepin observed throughout fermentation (see dynamic curve in Figure 7a) is likely linked to the growth phase of *C. militaris*, with peak production occurring during active mycelial proliferation, as similarly noted in previous submerged fermentation systems [32,33]. This confirms not only the metabolic compatibility of *C. militaris* with soy matrices but also the robustness of the strain under high-salt conditions.

Cordyceps polysaccharides were also detected in the CM group at a concentration of 35.44 mg/100 mL, whereas no polysaccharides were found in the control (Figure 7b). These polysaccharides, mostly β-glucans and galactomannans [34,35], have been associated with immunoenhancing properties in functional food applications, and their selective accumulation in CM soy sauce is linked to the unique biosynthetic capabilities of *C. militaris*. During the koji stage, as *C. militaris* mycelia proliferate, they specifically upregulate glucan synthases such as UDP-glucosyltransferases, which catalyze the linkage of glucose and mannose monomers (derived from the soy matrix) into β-glucans via glycosidic bonds—unlike *Aspergillus oryzae* in traditional fermentation, which primarily produces α-glucans in much lower quantities. This difference in glycosidic bond specificity explains why functional polysaccharides were exclusively detected in the CM sample. Additionally, *C. militaris* secretes high-activity cellulases and hemicellulases, whose production is likely enhanced by the optimized aeration and temperature conditions during koji preparation; these enzymes efficiently degrade soybean cell wall components such as cellulose and hemicellulose, releasing monosaccharides that serve not only as energy sources but also as precursors for polysaccharide synthesis, which are subsequently reassembled into functional polysaccharides via glycosyltransferases. Together, these two markers confirm that *C. militaris*-based fermentation significantly enhances the functional profile of soy sauce, opening new avenues for health-oriented condiments. These macromolecules are widely studied for immunostimulatory properties, and their presence in a daily-consumed seasoning product could offer functional food-level health benefits [36].

#### 3.3.2. Nutritional and Physicochemical Indices

As shown in Table 5, the CM soy sauce exhibited higher levels of amino acid nitrogen (Figure 8a) (1.14 ± 0.05 g/100 mL), total nitrogen (Figure 8b) (1.92 ± 0.04 g/100 mL), and salt-free soluble solids (Figure 8c) (22.17 ± 0.15 g/100 mL) compared to the TF soy sauce. These values not only exceed the national first-grade soy sauce standard (GB 18186-2000) but are also higher than the typical ranges reported for soy sauces fermented with *Aspergillus oryzae* alone (e.g., amino acid nitrogen 0.8–1.00 g/100 mL) [37,38], indicating more extensive proteolysis and nutrient release under *C. militaris*-dominated conditions. For instance, soy sauces fermented with *A. oryzae* have been reported to reach amino acid nitrogen levels up to 0.93 g/100 mL and total nitrogen around 1.59 g/100 mL after extended fermentation [27], which aligns with the typical range of 0.85–0.98 and 1.5–1.8 g/100 mL reported for conventional soy sauce, thus supporting the nutritional enhancement observed in CM soy sauce [1,39].

The total acid content of the CM soy sauce was 1.85 ± 0.08 g/100 mL (Figure 8d), slightly lower than that of the TF sample, suggesting a milder sourness profile. This could be attributed to the dominance of *C. militaris*, which does not produce lactic acid as extensively as lactic acid bacteria typically involved in traditional fermentations. Lactic acid bacteria such as *Tetragenococcus halophilus* are known contributors to total acid content in traditional soy sauce [40], which may explain the lower acid production in CM samples [41].

Color analysis (Figure 8e) revealed a darker tone in the CM sample, with reduced L* and elevated a* values, indicating an enhanced reddish-brown hue. This shift may be attributed to intensified Maillard reactions and higher levels of reducing sugars and free amino groups generated during fungal enzymatic hydrolysis, which may also contribute to the antioxidant capacity of the product [42].

#### 3.3.3. Volatile Flavor Compounds and Fatty Acid Profile

GC–MS analysis revealed a total of 57 volatile compounds across both soy sauce samples. These volatiles included acids, esters, aldehydes, ketones, phenols, alcohols, alkenes, hydrocarbons, and nitrogen-containing heterocycles. The compound profiles and relative abundances are summarized in Table 7.

The CM soy sauce exhibited a more lipid-centric volatile profile. Linoleic acid reached 15.68% in CM, nearly double that in TF (7.83%), significantly higher than the 7–10% range typically observed in soy sauces fermented with *Aspergillus oryzae* or *T. halophilus*, and consistent with enhanced lipid metabolism during *C. militaris* fermentation [3,43]. This observation aligns with the findings of Zou et al. and Chin et al., who reported linoleic acid as a key contributor to fermented aroma in Japanese soy sauce [44,45]. Fatty acid methyl esters such as methyl palmitate and methyl oleate were also elevated or uniquely detected in CM, indicating active esterification pathways [46,47]. *C. militaris* likely enhances the production of bioactive volatiles through a set of characteristic enzymatic pathways, as inferred from its metabolic traits reported in related studies. During fermentation, it may secrete lipoxygenases with strong activity toward polyunsaturated fatty acids (e.g., linoleic acid in soybeans), catalyzing oxidative reactions to generate aldehydes and ketones—consistent with the higher levels of hexanal and nonanal detected in CM soy sauce compared to traditional products. This contrasts with traditional fermentation strains, which tend to produce fewer such compounds due to differences in lipid-metabolizing enzyme profiles. In terms of ester synthesis, *C. militaris* may exhibit stronger esterification capacity, promoting reactions between fatty acids and alcohols to accumulate esters like methyl palmitate. This aligns with the elevated ester content observed in CM samples, as traditional strains generally show weaker activity in long-chain fatty acid esterification. Additionally, *C. militaris* might produce specific enzymes involved in terpene metabolism, facilitating the generation of terpenoids (e.g., trans-squalene) [48] that are absent in traditional soy sauce, possibly as a metabolic adaptation to environmental conditions like high salt. Concurrent with lipid metabolism, proteases secreted by *C. militaris* not only contribute to protein degradation and amino acid release but may also generate branched-chain amino acids, which can serve as precursors for volatile aldehydes (e.g., methylbutanal) through further metabolic reactions—a pathway less prominent in traditional fermentation systems. Together, these coordinated enzymatic activities, based on known metabolic characteristics of *C. militaris*, likely contribute to the unique bioactive volatile profile of CM soy sauce. In contrast, the TF soy sauce contained higher levels of saturated dicarboxylic acids, such as azelaic acid (17.55%), often associated with oxidative degradation of fatty acids.

In terms of compound classes, CM soy sauce contained 11 acids, 4 esters, 3 ketones, 3 aldehydes and phenols, 4 alcohols and alkenes, 1 furan/pyrrole, 6 hydrocarbons, and 5 other compounds, showing slightly broader chemical diversity than TF. Compared to the traditional profile dominated by phenols, pyrazines, and furans [49,50], the CM soy sauce displayed a profile enriched in lipid derivatives and fungal secondary metabolites.

Esters, known for fruity and floral notes, were present at 4.37% in CM, with linoleic acid methyl ester (2.28%) being the dominant ester. Esters like methyl linoleate have been previously identified as aroma enhancers in fungal-fermented foods [51,52], further supporting their sensory contribution in the CM sample. TF showed higher content of phthalate esters such as diethyl phthalate (2.49%). The presence of unique esters in CM contributes to a softer, fruitier aroma. Ketones and aldehydes, which provide sweet and spicy notes, were also found in balanced proportions. The CM group uniquely contained tetrahydro-beta-ionone and 3,4-dimethoxyphenylacetone, contributing to its floral and sweet characteristics. Though TF had slightly higher total ketone content, CM’s aldehydes were more prominent, giving a richer aromatic depth. Alcohols such as triacontanol and niacinamide provided distinct waxy and medicinal aromas to CM soy sauce, while TF showed common phenolic alcohols like lilac alcohol and furfuryl alcohol [47,53]. The detection of trans-squalene, 1-eicosene, and other alkenes in CM also suggests enhanced terpenoid-like metabolism. Pyrroles and furans, especially 2-acetylpyrrole, were significantly enriched in CM (1.04%), while 2-pentylfuran was absent, differentiating it from TF. CM lacked some of the classic burnt or roasted notes associated with traditional soy sauces, indicating a smoother, milder flavor profile.

Triacontanol (1.37%) and niacinamide (0.22%) were uniquely detected in the CM group, suggesting active secondary metabolite synthesis by *C. militaris*. Triacontanol is a plant-derived long-chain alcohol with known biostimulant and waxy aromatic properties [54], while niacinamide (vitamin B3 amide) points to de novo vitamin biosynthesis pathways. These compounds were absent from the TF group, supporting the functional divergence introduced by fungal fermentation.

Compared to conventional soy sauces dominated by furans, pyrazines, and phenols generated by *Aspergillus oryzae* and lactic acid bacteria [55,56], the CM soy sauce presented a shift toward lipid-derived volatiles and fungal metabolites. Similar patterns have been observed in *C. militaris*-based fermentations on non-soy substrates [48], where enhanced fatty acid elongation and alcohol biosynthesis pathways are activated under nutrient-rich, high-protein conditions. Mechanistically, this shift may stem from the unique enzyme system of *C. militaris*, including lipoxygenases, acetyltransferases, and alcohol dehydrogenases, which transform endogenous lipids and proteins into structurally diverse volatiles.

These findings demonstrate that incorporating *C. militaris* into soy sauce fermentation substantially alters the volatile landscape—enhancing not only aroma complexity but also introducing bioactive metabolites with potential health benefits. Such metabolic reprogramming expands the functional value and flavor diversity of fermented soy condiments.

#### 3.3.4. Taste Profile Analysis

Electronic tongue analysis results are shown in Figure 9, with detailed sensor values summarized below. The CM soy sauce exhibited notably improved sensory attributes compared to the traditionally fermented (TF) counterpart.

The bitterness and astringency of the CM sample were significantly reduced, registering 4.78 and 4.70, respectively, versus 6.98 and 8.24 in the TF group—a ~30–40% decrease indicating improved palatability. Such reduction may be attributed to lower accumulation of bitter peptides and polyphenolic oxidation products. Indeed, phenolic compounds are known contributors to bitterness and astringency in fermented foods, and minimizing their degradation has been shown to enhance soy sauce acceptability.

The umami response was slightly lower in CM (8.36) than in TF (8.89), despite CM showing higher amino acid nitrogen content (1.14 g/100 mL). This minor difference may relate to the non-glutamate umami amino acid composition or masking effects from other volatiles [57]. However, richness, representing umami depth, was significantly higher in CM (13.32 vs. 7.16), indicating a more complex, lingering taste structure, likely due to the coordinated action of free amino acids and Maillard-derived peptides [57]. Saltiness in CM was substantially higher (8.22 vs. 4.02), potentially reflecting higher sodium ion release from matrix degradation or less salt masking from bitter/astringent compounds. Aftertaste-A (13.32) and Aftertaste-B (12.07) scores in CM showed balanced persistence, indicating better flavor retention and finish. Overall, the CM soy sauce demonstrated a more harmonious sensory structure, characterized by enhanced flavor depth, smoother mouthfeel, and reduced off-notes [58].

These results collectively indicate that *C. militaris* fermentation not only enhances nutritional and functional components but also contributes to a more refined and consumer-preferred flavor profile, which may increase product competitiveness in the functional condiment market.

#### 3.3.5. Functional Value and Comparative Discussion

The results collectively demonstrate that soy sauce fermented with *C. militaris* possesses a superior functional profile, integrating high-quality protein degradation with bioactive metabolite accumulation. Unlike traditional soy sauces, which emphasize flavor and preservation, the CM soy sauce offers dual benefits: culinary value and potential health-promoting effects. This distinction stems from fundamental differences in metabolic networks between *C. militaris* and conventional strains like *Aspergillus oryzae*. Traditional soy sauce fermentation relies primarily on the digestive enzyme systems of *A. oryzae* (e.g., proteases and amylases), focusing on the synthesis of flavor compounds such as glutamic acid and succinic acid. In contrast, *C. militaris* features a “dual-function” metabolic network: first, it not only secretes proteases (with activity reaching 1872 U/g under optimized koji conditions) to degrade soybean proteins into amino acids—enhancing umami—but also channels part of these amino acids into secondary metabolism, where enzymes like adenosine deaminase catalyze the conversion of adenosine to cordycepin. Second, during the high-salt fermentation phase (12% brine), *C. militaris* likely maintains metabolic activity by activating stress-responsive genes (e.g., heat shock protein genes) and upregulating antioxidants such as superoxide dismutase, which mitigate oxidative degradation of functional compounds. In contrast, traditional strains exhibit suppressed metabolic activity under high-salt conditions, limiting their ability to accumulate such bioactive substances.

Several studies support the health relevance of cordycepin and Cordyceps polysaccharides. Researchers reported that cordycepin can modulate immune signaling pathways and reduce pro-inflammatory cytokine expression [59,60]. Gu et al. demonstrated the hypoglycemic and hepatoprotective properties of *C. militaris*-derived β-glucans [61,62]. Linoleic acid, abundant in the CM soy sauce, has been linked to reduced LDL cholesterol and improved lipid metabolism [63].

These functional attributes support the positioning of CM soy sauce as a next-generation fermented condiment with value beyond taste. Moreover, the absence of added extracts and the use of native *C. militaris* fermentation enhance its marketability and clean-label potential.

#### 3.3.6. Limitations and Future Perspectives

Notably, this study has limitations that warrant attention for both scientific rigor and practical translation. First, functional claims for CM soy sauce are based on in vitro metabolite profiles, lacking in vivo validation (e.g., animal or clinical studies) to confirm bioavailability and physiological effects of cordycepin, polysaccharides, and other bioactives. Second, sensory perception may vary across consumer demographics due to cultural and dietary differences, requiring targeted sensory panels to correlate specific volatiles with preference. Third, strain specificity of *C. militaris* remains underexplored—variations in metabolic pathways (e.g., regulatory mechanisms of cordycepin biosynthesis enzymes) across strains could alter functional outputs, highlighting the need for comparative studies on strain-specific metabolic logic.

Fourth, food safety, regulatory compliance, and consumer acceptance—critical for commercialization of novel fungal-fermented products—require deeper investigation. Safety considerations include long-term toxicological assessments of key metabolites (e.g., volatile thresholds, potential synergistic effects) and screening for unintended byproducts (e.g., mycotoxins under suboptimal conditions). Regulatory alignment with regional frameworks (e.g., EU Novel Food regulations and China’s GB 16740-2014 [64] for edible fungi) and transparent communication of natural enzymatic origins (to enhance consumer trust) are also essential.

Future research should address these gaps to strengthen the scientific basis and translational potential of CM soy sauce.

## 4. Conclusions

This study successfully established an optimized fermentation process for soy sauce brewed with *C. militaris*, integrating both koji preparation and liquid-state fermentation. Using response surface methodology, key parameters such as temperature, aeration, inoculum level, brine concentration, and water-to-material ratio were fine-tuned to enhance protease activity and maximize the accumulation of cordycepin and amino acid nitrogen. The resulting soy sauce demonstrated significantly higher levels of cordycepin (16.88 mg/100 mL), Cordyceps polysaccharides (35.44 mg/100 mL), and amino acid nitrogen (1.14 g/100 mL), surpassing traditional soy sauce in both nutritional and functional attributes. Unique volatile and lipid compounds further contributed to improved flavor and potential health benefits.

These findings highlight the feasibility of using *C. militaris* as a dominant strain in soy sauce fermentation, offering a novel strategy for developing next-generation functional condiments with dual culinary and bioactive value. Addressing the outlined limitations will accelerate its translation into commercial products.

## Figures and Tables

**Figure 1 foods-14-02711-f001:**
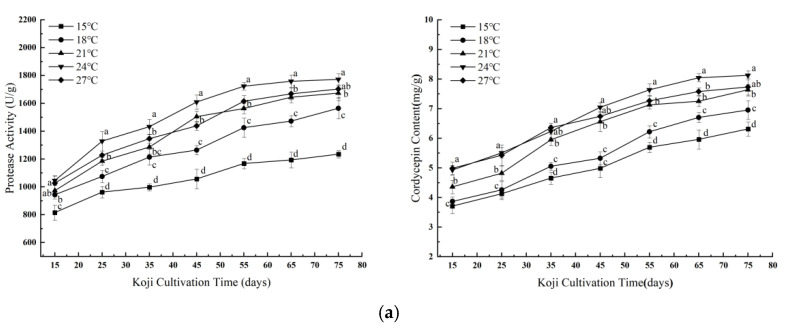
(**a**) Effect of Temperature on Protease Activity and Cordycepin Content. (**b**) Effect of aeration rate on protease activity and cordycepin content during koji preparation. (**c**) Effect of inoculum dosage on protease activity and cordycepin content during koji preparation. Values with different lowercase letters indicate statistically significant differences (*p* < 0.05, Duncan’s multiple range test).

**Figure 2 foods-14-02711-f002:**
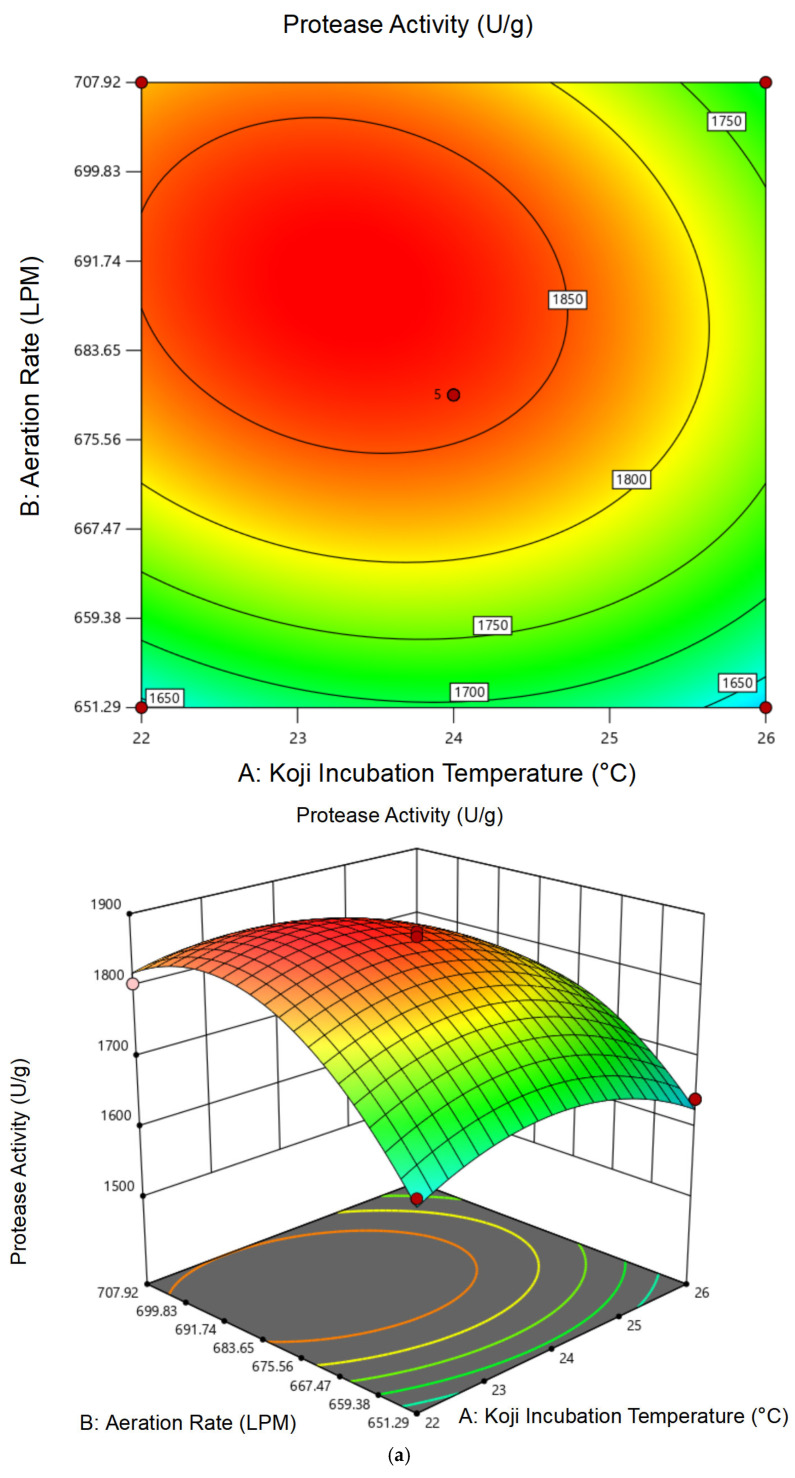
Response surface plots showing the interaction effects of process variables on protease activity during koji preparation: (**a**) temperature × aeration rate; (**b**) temperature × inoculum dosage; (**c**) aeration rate × inoculum dosage.

**Figure 3 foods-14-02711-f003:**
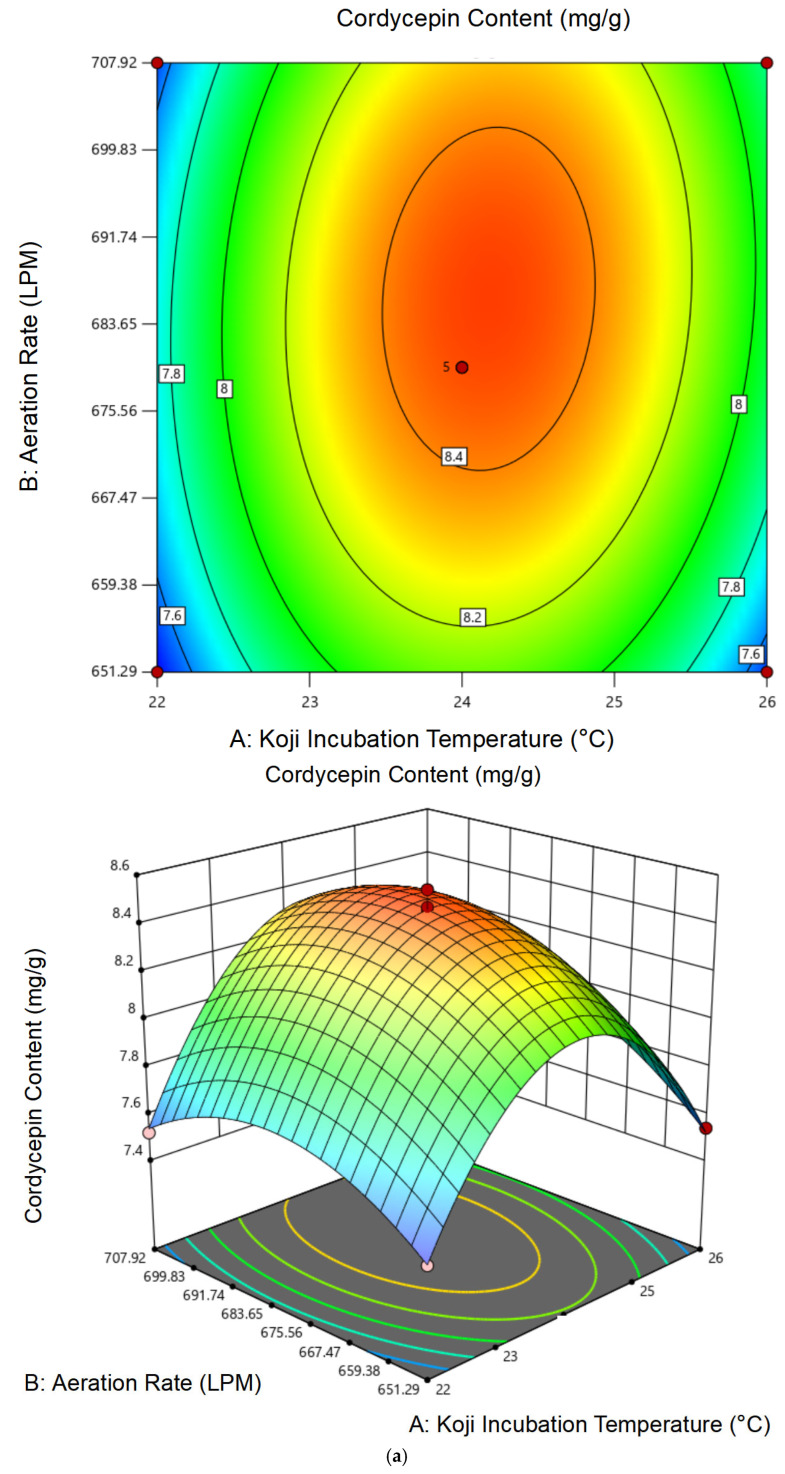
Response surface plots showing the interaction effects of process variables on cordycepin content during koji preparation: (**a**) temperature × aeration rate; (**b**) temperature × inoculum dosage; (**c**) aeration rate × inoculum dosage.

**Figure 4 foods-14-02711-f004:**
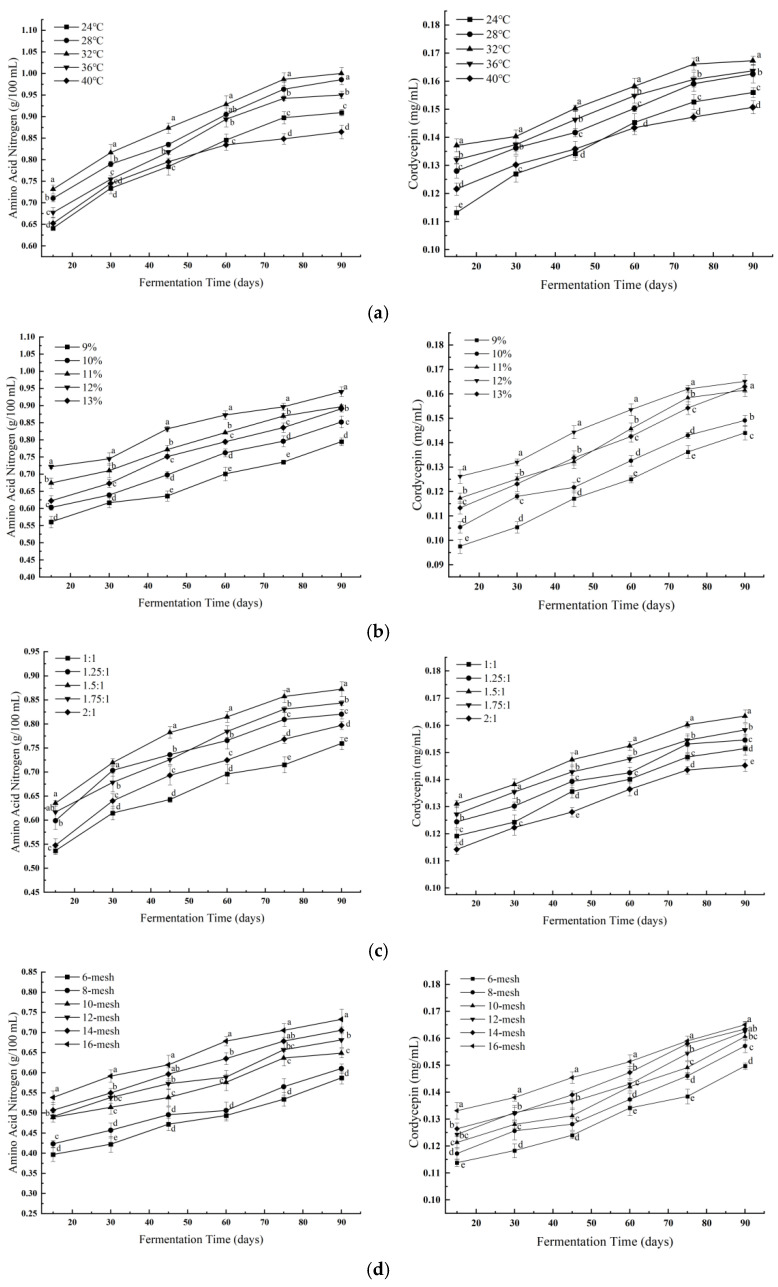
(**a**) Changes in amino acid nitrogen and cordycepin content during fermentation at different temperatures. (**b**) Changes in amino acid nitrogen and cordycepin content during fermentation at different brine concentrations. (**c**) Changes in amino acid nitrogen and cordycepin content during fermentation with varying brine volumes. (**d**) Changes in amino acid nitrogen and cordycepin content during fermentation with different koji particle sizes. Values with different lowercase letters indicate statistically significant differences (*p* < 0.05, Duncan’s multiple range test).

**Figure 5 foods-14-02711-f005:**
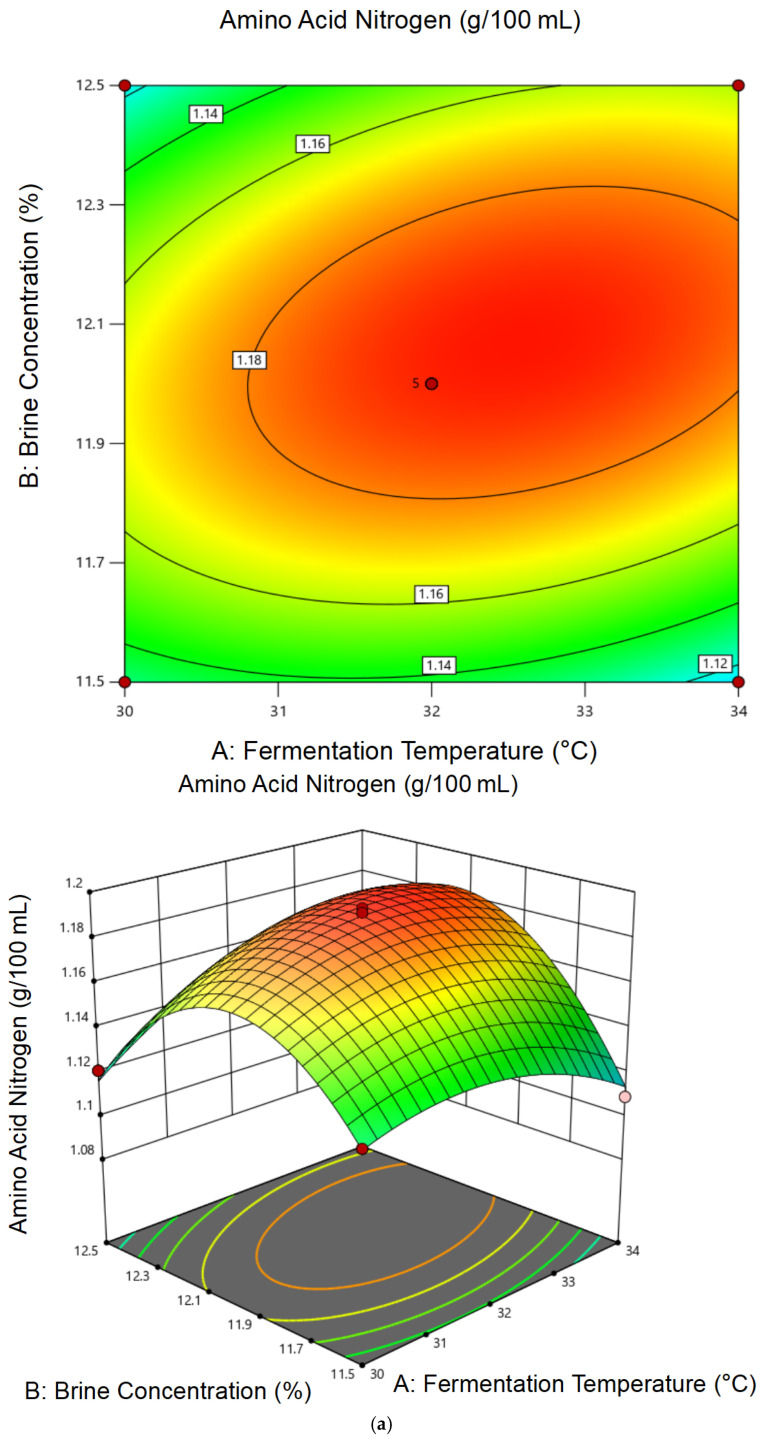
Response surface plots illustrating the interaction effects of fermentation parameters on amino acid nitrogen content: (**a**) temperature × brine concentration, (**b**) temperature × water-to-material ratio, and (**c**) brine concentration × water-to-material ratio.

**Figure 6 foods-14-02711-f006:**
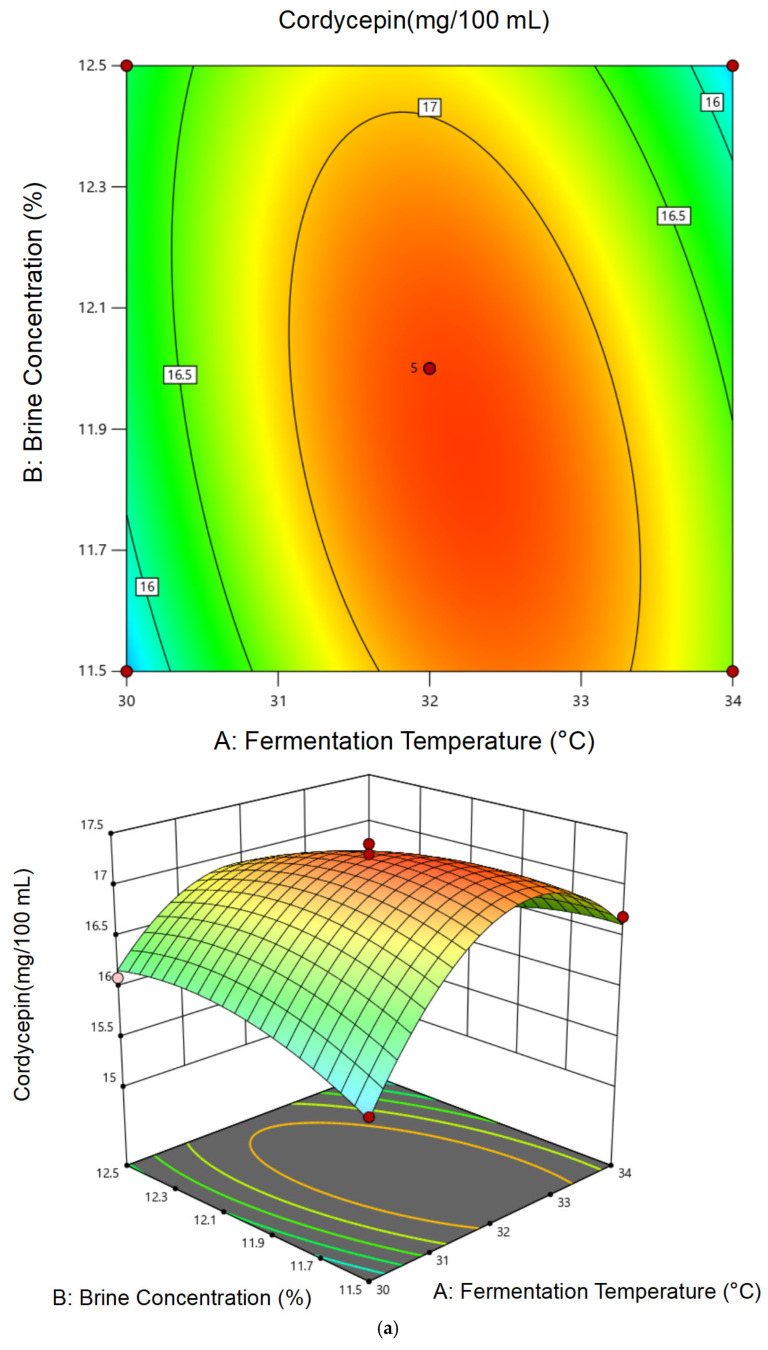
Response surface plots illustrating the interaction effects of key fermentation parameters on cordycepin content: (**a**) temperature × brine concentration, (**b**) temperature × water-to-material ratio, and (**c**) brine concentration × water-to-material ratio.

**Figure 7 foods-14-02711-f007:**
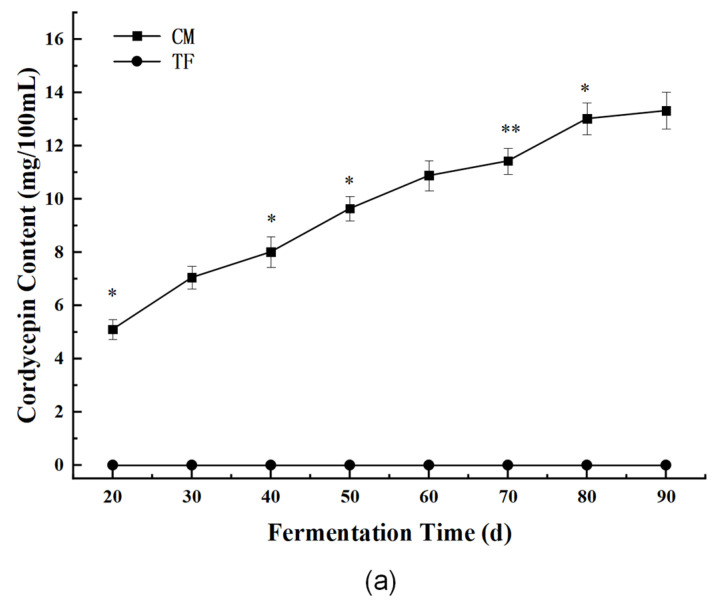
Changes in cordycepin (**a**) and polysaccharide (**b**) content during fermentation in the Cordyceps and control groups. * Asterisks denote statistical significance: * *p* < 0.05, ** *p* < 0.01.

**Figure 8 foods-14-02711-f008:**
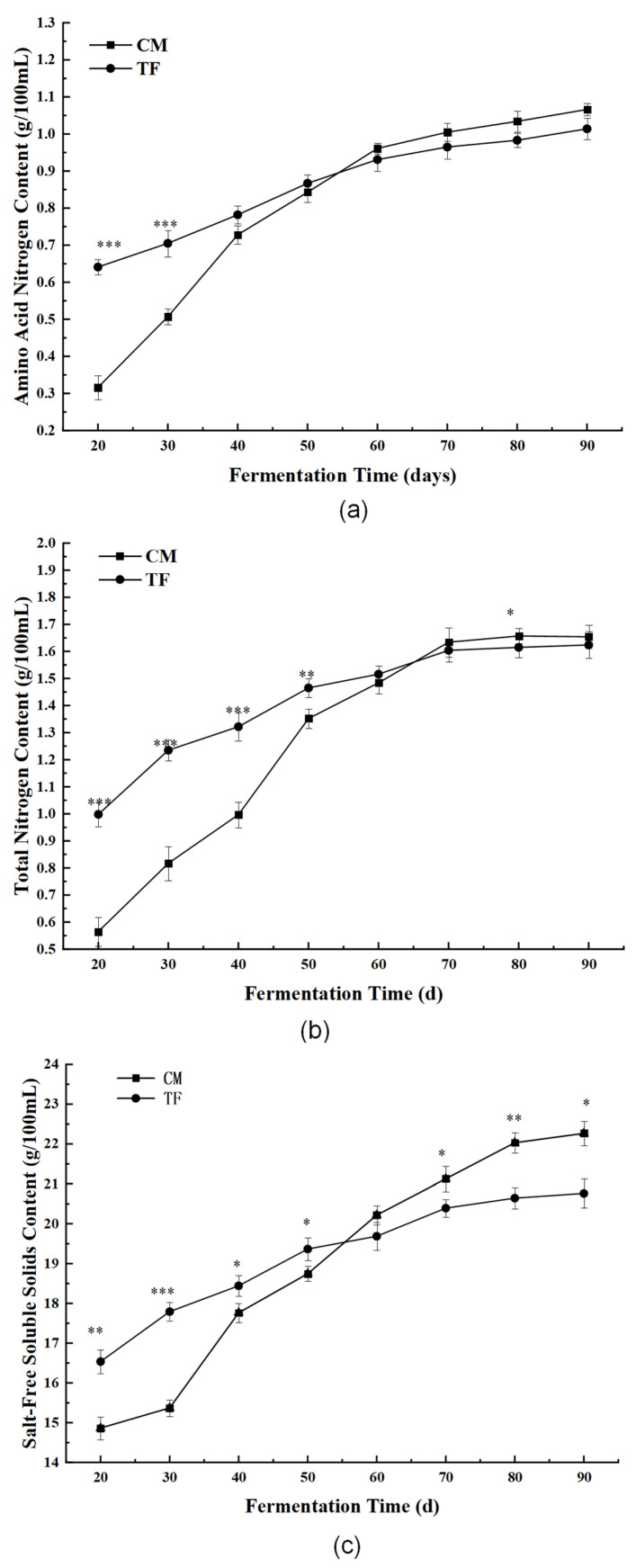
Comparison of key nutritional and physicochemical indices between traditionally fermented soy sauce (TF) and *Cordyceps militaris*-fermented soy sauce (CM). (**a**) Amino acid nitrogen; (**b**) total nitrogen; (**c**) salt-free soluble solids; (**d**) total acid content; and (**e**) color parameters (L*, a*, b*) based on the CIELAB system. Data are expressed as mean ± standard deviation (n = 3). * Asterisks denote statistical significance: * *p* < 0.05, ** *p* < 0.01, *** *p* < 0.001.

**Figure 9 foods-14-02711-f009:**
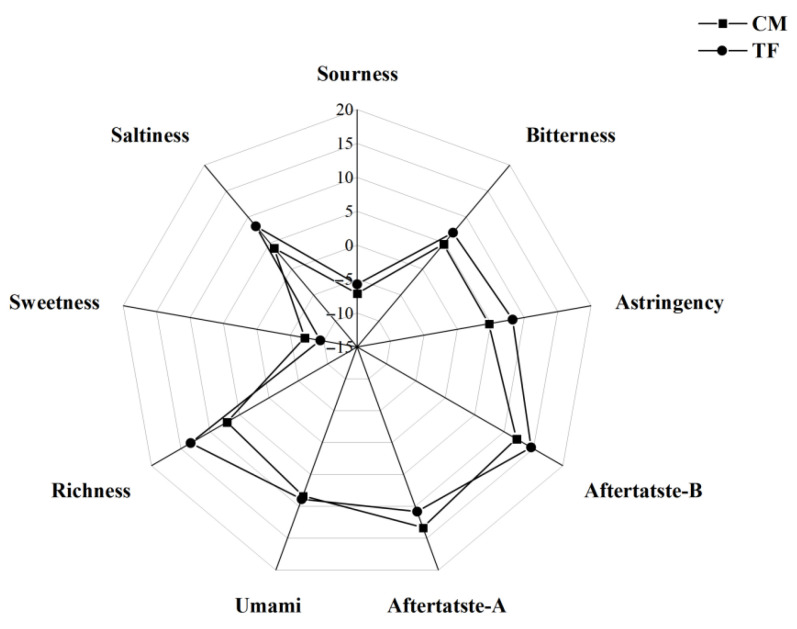
Radar chart of taste attributes of soy sauce samples.

**Table 1 foods-14-02711-t001:** Response surface test scheme and results.

Experiment No.	A	B	C	Protease Activity (U/g)	Cordycepin Content (mg/g)
1	24	23	8	1685.32	8.06
2	24	25	12	1817.24	8.11
3	22	23	10	1652.37	7.46
4	24	23	12	1562.8	7.95
5	26	25	10	1694.72	7.86
6	26	24	12	1617.58	7.69
7	24	25	8	1739.27	8.34
8	22	25	10	1804.23	7.52
9	24	24	10	1875.77	8.44
10	26	24	8	1801.13	7.94
11	24	24	10	1851.05	8.47
12	24	24	10	1847.51	8.39
13	24	24	10	1869.51	8.46
14	22	24	12	1826.84	7.57
15	22	24	8	1727.06	7.72
16	24	24	10	1868.61	8.54
17	26	23	10	1641.42	7.54

A represents temperature (22–26 °C), B represents aeration rate (651.29–707.92 LPM), and C represents inoculum concentration (8–12%).

**Table 2 foods-14-02711-t002:** Variance analysis of protease activity regression model.

Source of Variation	Sum of Squares	Degrees of Freedom	Mean Square	F-Value	*p*-Value	Significance
Model	1.584 × 10^5^	9	17,605.23	56.80	<0.0001	***
A	8169.62	1	8169.62	26.36	0.0013	**
B	32,966.19	1	32,966.19	106.35	<0.0001	***
C	2058.12	1	2058.12	6.64	0.0366	*
AB	2428.52	1	2428.52	7.83	0.0266	*
AC	20,068.97	1	20,068.97	64.75	<0.0001	***
BC	10,048.66	1	10,048.66	32.42	0.0007	***
A^2^	15,747.35	1	15,747.35	50.80	0.0002	***
B^2^	44,799.24	1	44,799.24	144.53	<0.0001	***
C^2^	14,253.24	1	14,253.24	45.98	0.0003	***
Residual	2169.75	7	309.96			
Lack of Fit	1551.39	9	517.13	3.35	0.1369	ns
Pure Error	618.37	1	154.59			
Total	1.606 × 10^5^	1				
R^2^	0.9865	1				
R^2^_Adj_	0.9691	1				

For definitions of A, B, and C, see Table 1. * Asterisks indicate statistical significance: * *p* < 0.05, ** *p* < 0.01, *** *p* < 0.001, ns = not significant.

**Table 3 foods-14-02711-t003:** Analysis of variance of regression model of cordycepin content.

Source of Variation	Sum of Squares	Degrees of Freedom	Mean Square	F-Value	*p*-Value	Significance
Model	2.31	9	0.2562	132.86	<0.0001	***
A	0.0722	1	0.0722	37.44	0.0005	***
B	0.0841	1	0.0841	43.58	0.0003	***
C	0.0685	1	0.0685	35.49	0.0006	***
AB	0.0169	1	0.0169	8.76	0.0211	*
AC	0.0025	1	0.0025	1.30	0.2924	ns
BC	0.0036	1	0.0036	1.87	0.2141	ns
A^2^	1.64	1	1.64	852.83	<0.0001	***
B^2^	0.2425	1	0.2425	125.75	<0.0001	***
C^2^	0.0464	1	0.0464	24.07	0.0017	**
Residual	0.0135	7	0.0019			
Lack of Fit	0.0017	3	0.0006	0.1921	0.8967	ns
Pure Error	0.0118	4	0.0029			
Total	2.32	16				
R^2^	0.9942					
R^2^_Adj_	0.9867					

For definitions of A, B, and C, see Table 1. * Asterisks indicate statistical significance: * *p* < 0.05, ** *p* < 0.01, *** *p* < 0.001, ns = not significant.

**Table 4 foods-14-02711-t004:** Response surface test design and results.

Run	A	B	C	Amino Acid Nitrogen (g/100 mL)	Cordycepin (mg/100 mL)
1	32	12.5	1.35:1	1.099	15.9
2	30	11.5	1.5:1	1.131	15.7
3	30	12	1.65:1	1.165	15.8
4	32	12	1.5:1	1.193	17.4
5	34	12	1.65:1	1.167	15.8
6	34	12	1.35:1	1.142	15.7
7	32	12.5	1.65:1	1.147	16.7
8	30	12.5	1.5:1	1.121	16.1
9	32	12	1.5:1	1.186	17.2
10	30	12	1.35:1	1.107	15.3
11	32	11.5	1.35:1	1.091	16.2
12	32	12	1.5:1	1.191	17.2
13	34	12.5	1.5:1	1.159	15.7
14	32	11.5	1.65:1	1.131	16.6
15	34	11.5	1.5:1	1.109	16.7
16	32	12	1.5:1	1.191	17.3
17	32	12	1.5:1	1.187	17.2

A represents fermentation temperature (30, 32, 34 °C); B represents brine concentration (11.5%, 12.0%, 12.5%); C represents water-to-material ratio (1.35:1, 1.50:1, 1.65:1).

**Table 5 foods-14-02711-t005:** Analysis of variance of regression model of amino acid nitrogen content.

Source of Variation	Sum of Squares	Degrees of Freedom	Mean Square	F-Value	*p*-Value	Significance
Model	0.0195	9	0.0022	121.08	<0.0001	***
A	0.0004	1	0.0004	19.59	0.0031	**
B	0.0005	1	0.0005	28.57	0.0011	**
C	0.0037	1	0.0037	203.95	<0.0001	***
AB	0.0009	1	0.0009	50.22	0.0002	***
AC	0.0003	1	0.0003	15.19	0.0059	**
BC	0.0000	1	0.0000	0.8928	0.3762	ns
A^2^	0.0010	1	0.0010	57.73	0.0001	***
B^2^	0.0081	1	0.0081	453.30	<0.0001	***
C^2^	0.0035	1	0.0035	193.18	<0.0001	***
Residual	0.0001	7	0.0000			
Lack of Fit	0.0001	3	0.0000	3.42	0.1330	ns
Pure Error	0.0000	4	8.800 × 10^−6^			
Total	0.0197	16				
R^2^	0.9936					
R^2^_Adj_	0.9854					

For definitions of A, B, and C, see Table 1. * Asterisks indicate statistical significance: ** *p* < 0.01, *** *p* < 0.001, ns = not significant.

**Table 6 foods-14-02711-t006:** Analysis of variance of regression model of cordycepin content.

Source of Variation	Sum of Squares	Degrees of Freedom	Mean Square	F-Value	*p*-Value	Significance
Model	7.62	9	0.8470	58.13	<0.0001	***
A	0.1250	1	0.1250	8.58	0.0221	*
B	0.0800	1	0.0800	5.49	0.0516	ns
C	0.4050	1	0.4050	27.79	0.0012	**
AB	0.4900	1	0.4900	33.63	0.0007	***
AC	0.0400	1	0.0400	2.75	0.1415	ns
BC	0.0400	1	0.0400	2.75	0.1415	ns
A^2^	3.84	1	3.84	263.54	<0.0001	***
B^2^	0.2738	1	0.2738	18.79	0.0034	**
C^2^	1.81	1	1.81	123.97	<0.0001	***
Residual	0.1020	7	0.0146			
Lack of Fit	0.0700	3	0.0233	2.92	0.1639	ns
Pure Error	0.0320	4	0.0080			
Total	7.72	16				
R^2^	0.9868					
R^2^_Adj_	0.9698					

For definitions of A, B, and C, see Table 1. * Asterisks indicate statistical significance: * *p* < 0.05, ** *p* < 0.01, *** *p* < 0.001, ns = not significant.

**Table 7 foods-14-02711-t007:** Qualitative analysis of soy sauce flavor compounds.

No.	Compound	Molecular Formula	Molecular Weight	TF Relative Content (%)	CM Relative Content (%)	Significance
1	Furfural Alcohol	C_5_H_6_O_2_	98.037	0.31 ± 0.061	ND	***
2	Phenol	C_6_H_6_O	94.042	0.24 ± 0.014	ND	***
3	2-n-Pentylfuran	C_9_H_14_O	138.104	0.08 ± 0.006	ND	***
4	n-Hexanoic Acid	C_6_H_12_O_2_	116.084	0.16 ± 0.009	0.04 ± 0.002	**
5	2,3-Dimethyl-5-hydroxy-2-cyclopenten-1-one	C_7_H_10_O_2_	126.068	0.11 ± 0.006	ND	***
6	2-Acetylpyrrole	C_6_H_7_NO	109.053	0.86 ± 0.053	1.04 ± 0.103	*
7	4-Methyl-5,6-dihydro-2H-pyran-2-one	C_6_H_8_O_2_	112.052	0.32 ± 0.019	ND	***
8	Benzoic Acid	C_7_H_6_O_2_	122.037	1.12 ± 0.093	1.18 ± 0.071	ns
9	Azelaic Acid	C_9_H_16_O_4_	188.105	17.55 ± 1.013	ND	***
10	Phenylacetic Acid	C_8_H_8_O_2_	136.052	1.68 ± 0.095	1.39 ± 0.067	**
11	2,6-Dimethoxyphenol (Syringol)	C_8_H_10_O_3_	154.063	0.67 ± 0.047	0.64 ± 0.043	ns
12	2,6-Di-tert-butyl-p-cresol (BHT)	C_15_H_24_O	220.183	2.7 ± 0.170	7.71 ± 0.458	***
13	Diethyl Phthalate	C_12_H_14_O_4_	222.089	2.49 ± 0.217	0.46 ± 0.027	***
14	Cycloundecanone	C_11_H_20_O	168.151	0.57 ± 0.033	ND	***
15	2,4-Di-tert-butylphenol	C_14_H_22_O	206.167	0.27 ± 0.016	ND	***
16	Methyl Linoleate	C_19_H_34_O_2_	294.256	1.09 ± 0.063	2.28 ± 0.004	***
17	3-Hydroxy-4-methoxybenzoic Acid	C_8_H_8_O_4_	168.042	0.5 ± 0.029	1.25 ± 0.109	***
18	Vanillic Acid	C_8_H_8_O_4_	168.042	0.96 ± 0.055	0.76 ± 0.044	**
19	Myristic Acid	C_14_H_28_O_2_	228.209	4.2 ± 0.243	ND	***
20	2-Octylcyclopropaneoctanal	C_19_H_36_O	280.277	0.6 ± 0.042	ND	***
21	Dehydroacetic Acid	C_8_H_8_O_4_	168.042	0.2 ± 0.011	ND	***
22	Oleamide	C_18_H_35_NO	281.272	0.48 ± 0.028	1.03 ± 0.009	***
23	Palmitic Acid	C_16_H_32_O_2_	256.24	5.59 ± 0.436	ND	***
24	Methyl Oleate	C_19_H_36_O_2_	296.272	0.79 ± 0.000017	ND	***
25	Petroselinic Acid	C_18_H_34_O_2_	282.256	0.04 ± 0.002	ND	***
26	Linoleic Acid	C_18_H_32_O_2_	280.24	7.83 ± 0.28	15.68 ± 0.132	***
27	2-Methylheptadecane	C_15_H_28_	208.219	0.11 ± 0.006	ND	***
28	Bis(2-ethylhexyl) Adipate	C_22_H_42_O_4_	370.308	0.72 ± 0.043	ND	***
29	(9Z)-Octadeca-9,17-dienal	C_18_H_32_O	264.245	0.54 ± 0.031	1.27 ± 0.122	***
30	Di(2-ethylhexyl) Phthalate	C_24_H_38_O_4_	390.277	0.75 ± 0.044	ND	***
31	Hexahydro-3-(phenylmethyl)-1H-pyrrolo [1,2-a]pyrazine-1,4-dione	C_14_H_16_N_2_O_2_	244.121	3.06 ± 0.107	1.9 ± 0.034	***
32	trans-13-Octadecenoic Acid	C_18_H_34_O_2_	282.256	0.28 ± 0.016	0.08 ± 0.146	***
33	2-Chloroethyl Linoleate	C_20_H_35_ClO_2_	342.233	0.05 ± 0.003	ND	***
34	trans-Squalene	C_30_H_50_	410.391	0.36 ± 0.014	0.69 ± 0.172	*
35	2-Hydroxycyclopentadecanone	C_15_H_28_O_2_	240.209	0.3 ± 0.001	0.4 ± 0.0492	*
36	Glyceryl Monooleate	C_21_H_40_O_4_	356.293	0.01 ± 0.082	ND	ns
37	Acetylpropionic Acid	C_8_H_16_O_2_	144.115	ND	0.42 ± 0.046	***
38	Nicotinamide	C_6_H_6_N_2_O	122.048	ND	0.22 ± 0.014	***
39	Phenylmalonic Acid	C_9_H_8_O_4_	180.042	ND	0.03 ± 0.002	ns
40	3,4-Dimethoxyphenylacetone	C_10_H_12_O_3_	180.079	ND	0.24 ± 0.014	***
41	Dehydroacetic Acid	C_8_H_8_O_4_	168.042	ND	0.51 ± 0.029	***
42	Triacontanol	C_30_H_62_O	438.48	ND	1.37 ± 0.079	***
43	Tetrahydro-α-ionone	C_13_H_24_O	196.183	ND	0.47 ± 0.0001	***
44	Oleamide	C_18_H_35_NO	281.272	ND	1.03 ± 0.009	***
45	Methyl Palmitate	C_17_H_34_O_2_	270.256	ND	1.04 ± 0.132	***
46	n-Nonacosane	C_29_H_60_	408.47	ND	0.34 ± 0.030	***
47	Cyclotriacontane	C_30_H_52_O_4_	420.47	ND	0.15 ± 0.023	***
48	Cyclo(L-Leu-L-Leu)	C_12_H_22_N_2_O_2_	226.168	ND	0.52 ± 0.074	***
49	(Z)-13-Octadecenal	C_18_H_34_O	266.261	ND	2.52 ± 0.013	***
50	n-Hexacosane	C_26_H_54_	366.423	ND	2.29 ± 0.018	***
51	Eicosane	C_20_H_42_	282.329	ND	0.22 ± 0.007	***
52	13-Tetradecen-1-ol Acetate	C_16_H_30_O_2_	254.225	ND	0.59 ± 0.035	***
53	n-Tricosane	C_23_H_48_	324.376	ND	0.31 ± 0.024	***
54	Tetracosane	C_24_H_50_	338.391	ND	0.12 ± 0.013	***
55	1-Eicosene	C_20_H_40_	280.313	ND	0.32 ± 0.015	***
56	Erucamide	C_22_H_43_NO	337.334	ND	0.23 ± 0.012	***
57	Dodecenyl Succinic Anhydride	C_16_H_26_O_3_	266.188	ND	1.91 ± 0.032	***

ND values were substituted with zero for statistical analysis where applicable. * Asterisks indicate statistical significance: * *p* < 0.05, ** *p* < 0.01, *** *p* < 0.001, ns = not significant.

## Data Availability

The original contributions presented in this study are included in the article. Further inquiries can be directed to the corresponding author.

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
