# Peer review of "Soy Sauce Fermentation with Cordyceps militaris: Process Optimization and Functional Profiling"

_foods, 2025, doi:10.3390/foods14152711_

Round 1

Reviewer 1 Report

Comments and Suggestions for Authors

The manuscript “Soy Sauce Fermentation with Cordyceps militaris: Process Optimization and Functional Profiling” presents an innovative and well-structured study on the fermentation of soy sauce using Cordyceps militaris as the sole microbial starter. The research is timely, methodologically sound, and addresses a relevant topic in the development of functional foods. The manuscript demonstrates high scientific quality and potential for publication, subject to some important revisions.

The study integrates microbiology, food chemistry, and sensory science in a coherent manner. Both the koji preparation and fermentation optimization are designed using statistically robust response surface methodology (Box–Behnken design). Analytical methods are well-described and follow standard protocols (e.g., HPLC, GC–MS, electronic tongue). Comparative analysis with traditionally fermented soy sauce strengthens the applied relevance of the findings.

While results is comprehensive, there is no clear “Limitations and Future Perspectives” section. This is highly recommended, especially considering that the strain specificity of C. militaris and potential regulatory or safety implications are not discussed.

Consider adding a short paragraph in the Discussion or Conclusion acknowledging:
Lack of in vivo validation of functional properties.
Possible sensory variations across demographics.
The need for shelf-life and stability studies in future research.

Many of the figures (especially Figures 2, 3, 5, 6, 7, and 8) are pixelated or lack numerical clarity. This affects readability, particularly in the 3D surface plots and radar charts.Figures 7 and 8 are critically low in resolution. Values are not legible. 

For surface response plots, consider splitting them into individual figures or increasing size in layout.

Include scale bars, larger fonts, and color contrast for enhanced clarity.

The name Cordyceps militaris should be italicized throughout the text. Although C. militaris is correctly abbreviated, please ensure first mention in each section spells out the full name in italics, followed by abbreviation in parentheses.

A few sentences are slightly long or repetitive. Consider shortening for clarity (e.g., lines 295–296, 413–414).

Ensure consistent usage of units (e.g., sometimes written as "g/100 mL", other times as “g·100⁻¹ mL”).

Although comparative results are included, there is little mechanistic speculation about why C. militaris enhances certain functional compounds.

Please expand the discussion on potential molecular or enzymatic pathways (e.g., lipoxygenases, proteases) involved in generating bioactive volatiles.

No mention is made of food safety, regulatory considerations, or consumer acceptance for novel fungal fermented products. A brief note in the conclusion or future perspectives would be appropriate.

Reviewer 2 Report

Comments and Suggestions for Authors

This MS is well-written, and the experimental designs are sound, but need some minor revisions.

Below are suggested changes:

  • Please convert CFM to lpm (litres per minute) as standard unit throughout the MS.
  • Please change “ventilation” to aeration throughout the MS.
  • It is also important to indicate how aeration was provided – was it in the incubator or surface of inoculated substrate, etc.?

Round 2

Reviewer 1 Report

Comments and Suggestions for Authors

The authors have addressed all the corrections I suggested and submitted their responses appropriately. Although the changes were not highlighted in the revised manuscript, I was able to identify them. I recommend that, in future submissions, the authors highlight all modifications to facilitate the review process. Nevertheless, this does not diminish the scientific merit of the manuscript, which I consider suitable for publication in its current revised form.